# OST: Improving Generalization of DeepFake Detection via One-Shot Test-Time Training

Liang Chen[1]    Yong Zhang[2*]    Yibing Song[2]    Jue Wang[2]    Lingqiao Liu[1*]
[1] The University of Adelaide    [2] Tencent AI Lab
{liangchen527, zhangyong201303, yibingsong.cv, arphid}@gmail.com
lingqiao.liu@adelaide.edu.au

## Abstract

State-of-the-art deepfake detectors perform well in identifying forgeries when they are evaluated on a test set similar to the training set, but struggle to maintain good performance when the test forgeries exhibit different characteristics from the training images, e.g., forgeries are created by unseen deepfake methods. Such a weak generalization capability hinders the applicability of current deepfake detectors. In this paper, we introduce a new learning paradigm specially designed for the generalizable deepfake detection task. Our key idea is to construct a test-sample-specific auxiliary task to update the model before applying it to the sample. Specifically, we synthesize pseudo-training samples from each test image and create a test-time training objective to update the model. Moreover, we propose to leverage meta-learning to ensure that a fast single-step test-time gradient descent, dubbed one-shot test-time training (OST), can be sufficient for good deepfake detection performance. Extensive results across several benchmark datasets demonstrate that our approach performs favorably against existing arts in terms of generalization to unseen data and robustness to different post-processing steps.

## 1   Introduction

Deep neural network models have brought remarkable advances to image editing and generation techniques. While such progress is revolutionizing the multimedia production industry, it also creates negative social impacts since it has never been easier to create highly deceivable forgery images. Among those new technologies, deepfake, which uses deep learning models to substitute the identity of one person with another or alter the facial features in a portrait, is particularly harmful since it can lead to severe digital crime and undermine the social trust system. To counteract such a negative impact, deepfake detection technique is developed to automatically recognize pristine or forgery and is receiving increasing attention in the research community [28, 35, 57, 23, 42, 4, 39, 58, 16, 24, 22, 9].

So far, existing deepfake detection methods achieve favorable performances when training and test forgeries are from the same dataset and generated by the same deepfake method. In practice, the test forgeries are usually generated by unknown methods or applied with different image postprocessing approaches. This discrepancy will inevitably create a distribution drift between training and test data. Unfortunately, existing deepfake detectors do not generalize well, and their performance tends to decrease significantly when evaluated across datasets. This phenomenon inspires recent studies [28, 42, 35, 57, 39] on improving model generalizations to recognize face forgeries generated from unseen methods. For example, a two heads network is proposed in [28] to discriminate face forgeries by amplifying blending artifacts. In [42], a face forgery frequency network is developed to mine forgery patterns in frequency domains for better generalization. Detail discrepancies are investigated

---

*Corresponding authors. This work is done when L. Chen is an intern in Tencent AI Lab. Code is available at https://github.com/liangchen527/OST.

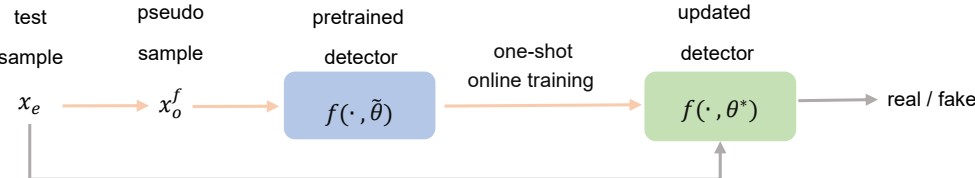

Figure 1: A glimpse of the proposed one-shot test-time training framework during online predictions. For every test sample $x_e$, we first synthesize a pseudo-training sample $x_o^f$ based on $x_e$. Then the pretrained detector can be updated via a supervised learning step with $x_o^f$, *i.e.* $x_o^f$ is with a known label as fake. The final result is obtained by applying the updated detector to the test sample.

in [35], and facial source feature inconsistencies are discovered in [57] to detect face forgeries, respectively. These methods try to explore common features among the training forgeries for better classification. But the test data often exhibit different characteristics, and the learned common features may not be shared by them. Since the test samples are not seen in the training stages, obtaining good generalization seems unreachable for current detectors.

This work introduces a new learning paradigm specially-designed for the generalizable deepfake detection task. Specifically, we allow the detector to "see" the test samples before making the final prediction by conducting an additional "training step" at the test time. One challenge of this idea is that the label of the test image is unavailable for the training objective. We overcome this problem by synthesizing a pseudo-training sample based on the test image and using it to update the deepfake detection model online. In a common deepfake detection setting [28, 57], no matter real or forgery for the evaluated test sample, we are confident that the synthesized pseudo sample is a forgery. This unique property enables our detector to train on the synthesized sample that has similar content to the test sample, thus better adapting to the test characteristic.

Moreover, we propose to use only one-step gradient descent, dubbed one-shot test-time training (OST), to update the model online for better computational efficiency. A glimpse of OST is shown in Figure 1. To ensure that such an OST scheme can always lead to a decent model without overfitting the pseudo-training sample, we employ meta-learning to train a good initial model in a similar style as MAML [19]. Note that although test-time training (TTT) has been previously proposed for the general image classification task [46], our approach adopts a different TTT objective that is more specific for deepfake detection. Meanwhile, as both suggested by a recent work [34] and our experimental study, the general-purposed self-supervised TTT algorithm in [46] may fail to bring any improvements but could even deteriorate the detection accuracy. On the contrary, our OST method can significantly improve the generalization performance of the deepfake detector and could attain superior performances over the existing solutions in multiple benchmark datasets.

## 2 Related Works

In this section, we conduct a brief survey on the most relevant arts, including existing deepfake detection methods and test-time training (TTT)-based works.

### 2.1 Deepfake Detection

Since the deepfake forgeries have led to great threats to societal security, it is of paramount importance to develop effective detectors against it. By formulating the detecting as a vanilla binary classification problem (*i.e.* pristine or forgery), current end-to-end trained detectors [43] with a simple Xception baseline [11] can obtain a high detection accuracy. Besides, with more powerful network structures and more informative image features embedded in the network inputs, existing methods [3, 2, 53, 36, 42, 17, 6, 39] are able to achieve even more remarkable success when the training and test forgeries are synthesized by the same deepfake algorithms. Thus, the real challenge in this task lies in how to generalize a learned detector to forgeries created by unseen methods.

Several works have been devoted to addressing the generalizing problem recently [28, 33, 7, 9, 8, 24]. For example, [28] suggests that the blending operation is ubiquitous in the current deepfake synthesizing process. As a consequence, they propose to detect the blending boundaries hidden in the forgeries and use them as classification clues. Moreover, [33] shows that the up-sampling

step in synthesizing models can bring artifacts to the synthesized forgeries, and they use the phase spectrums of the forgeries to capture these artifacts. As existing forgery synthesizing steps often involve two images from different identities and different sources, [35] suggest using high-pass filters from SRM [20] to reveal detail discrepancies of the forgeries. A similar idea is adopted in [57], where they use the cue of the source feature inconsistency within the forged images for detection. Although these methods are effective in many cases, the low-level artifacts they rely on are sensitive to post-processing steps that vary in different datasets, thus jeopardizing their generalization. Some other works propose to borrow features from other tasks, such as lips reading [23], facial image decomposition [60], and landmark geometric [47], to imply the abnormity of forgeries. Although these features can bring certain improvements, there is a great chance that future deepfake algorithms will be designed based on these detectors to synthesize more natural forgeries, causing even bigger threats to societal security.

Compared to existing detectors, the advantages of our method are as follows: (1) we adopt a MAML-based OST framework to enable the fast adaptation of the learned detector to the test data, which improves generalization regardless of the varying post-processing steps; (2) OST does not rely on hand-crafted or borrowed features, which leaves fewer traces for the deepfake algorithms to attack.

## 2.2 Test-time Training

The concept of TTT was firstly introduced in [46] for generalization to out-of-distribution test data, where a self-supervised rotation prediction task is utilized with the main classification task during training, and only the self-supervised task is adopted to help improve the visual representation during inference, which indirectly improves semantic classification. This framework is theoretically proved to be effective [46] and is further used in other related areas [30, 51, 41, 5, 55]. For example, [30] proposes a reconstruction task within the main pose estimation framework, which can be trained by comparing the reconstructed image with the ground truth borrowed from other frames. [51] demonstrate that the predictions with lower entropy have lower error rates, and they use entropy to provide finetuning signals when given a test image. Instead of only minimizing the entropy of the predicted posterior, [41] also suggests maximizing the noise robustness of the feature representation during test. In some recent works [5, 55], the TTT framework has also been utilized within a model agnostic meta-learning (MAML) paradigm [19] which allows the trained model to be optimized in a way such that it can quickly adapt to any test images. To enable fast adaptation, these works use contrastive loss [5] or smooth loss [55] to finetune the models during the meta-test phase.

However, despite some encouraging results, current TTT methods aim to select empirical self-supervised tasks, which is at high risk of deteriorating the performance when the tasks are not properly selected [34]. This work introduces a new learning paradigm specially designed for the deepfake detection task. Recall that a forgery can be easily synthesized by blending two different images [28, 57]. We thus use a newly synthesized forgery as a pseudo-training sample to finetune the pretrained detector during inference. Our method is easy to implement and can avoid the tedious work of selecting an effective self-supervised task.

## 3 Proposed Method

Our method consists of an offline meta-training step and an online test-time training step. The online test-time-training step will first generate a pseudo-training sample for each test image and then perform a single gradient descent update to obtain the sample-specific model parameters. The offline meta-training mimics the online test-time-training operation by sampling episodes from training data. In the following, we first describe the test-time-training step, involving both pseudo-training sample creation and one-shot update. Then we elaborate on the meta-learning process.

### 3.1 Online Test-Time Training

In the following, we assume that a deepfake detection model $\tilde{\theta}$ has already been learned. The OST process will create an updated model parameter $\theta^*$ that is adaptive to each individual test image. This is achieved by first generating a pseudo-training sample and then using it to form a mini-training set and update model parameters via one-shot training.

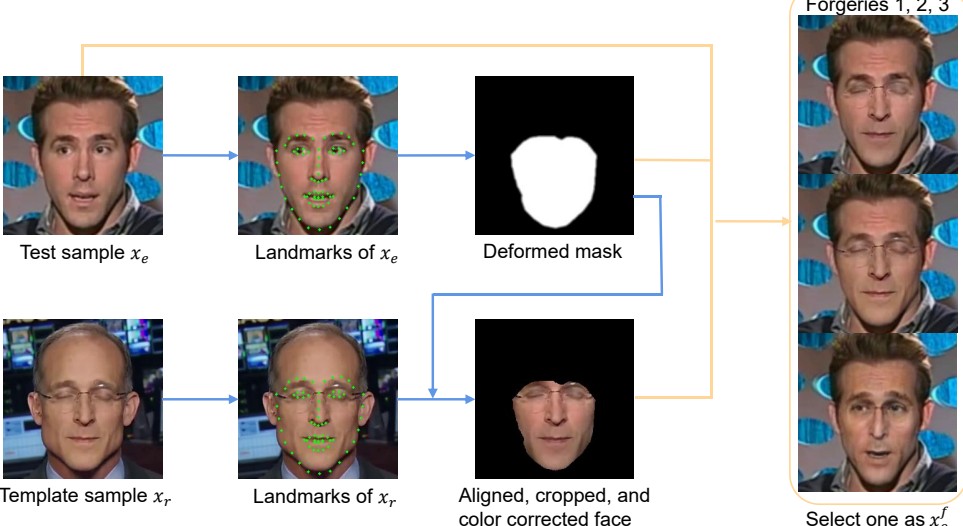

Figure 2: Pipeline for generating pseudo training samples. Forgeries 1, 2, 3 (from top to bottom) in the right are produced by the alpha, Poisson [40], and learning-based blending [10] methods, respectively.

---

**Algorithm 1** One-shot online training

---

**Require:** Pretrained detector $f(\cdot, \tilde{\theta})$, learning rate $\gamma$, test sample $x_e$, training dataset $\mathbf{D}$.

1: Ramdomly select $x_r$ from $\mathbf{D}$, and synthesize forgery $x_o^f$ by blending $x_e$ and $x_r$

2: Evaluate the one-shot loss $\displaystyle\sum_{x_s \in \{x_o^f, x_r\}} L(f(x_s, \tilde{\theta}), y_s)$

3: Update parameters with gradient descent: $\theta^* = \tilde{\theta} - \gamma \displaystyle\sum_{x_s \in \{x_o^f, x_r\}} \nabla_{\tilde{\theta}} L(f(x_s, \tilde{\theta}), y_s)$

4: Get the final classification result by evaluating the updated detector $f(\cdot, \theta^*)$ in the test sample $x_e$

---

**Generating pseudo-training samples:** As shown in Figure 2, For every test sample $x_e$, we first randomly select a template image $x_r$ from the training dataset and align these two images in geometry based on their landmarks. Then, we can extract a convex hull using the landmarks of $x_e$ and obtain the deformed final mask by applying random deformation and blurring on the convex hull. With the deformed mask, we can crop the corresponding color corrected face content from $x_r$. Finally, with $x_e$, the deformed mask, and the cropped face, we can obtain the pseudo training sample $x_o^f$ by using alpha or Poisson [40] blending methods as shown in the right side of Figure 2. Note that when using the leaning-based blending [10] method, we can directly use $x_r$ and $x_e$ as input and output $x_o^f$, which does not require the alignment, cropping, or correction steps. In the following OST step, we randomly select a forgery as $x_o^f$. These pseudo-training samples are labeled as negative (i.e., fake). Note that although we are unknown about whether the test image is pristine or forgery (i.e., real or fake), we are confident that the synthesized pseudo image is a forgery.

**One-shot online training:** In practice, we use the images with known labels (i.e., the randomly sampled template image $x_r$ and the synthesized image $x_o^f$) to form a mini-training set. The label of $x_r$ is provided by the training set, e.g., it can be real or fake dependent on the sampled image, while $x_o^f$ is always fake since it is a synthesized image. Then we can perform gradient descent to update the model. In our design, we only perform a single step gradient descent, dubbed one-shot online training. Formally, the initial model parameter is updated via

$$\theta^* = \tilde{\theta} - \gamma \sum_{x_s \in \{x_o^f, x_r\}} \nabla_{\tilde{\theta}} \mathcal{L}(f(x_s, \tilde{\theta}), y_s), \tag{1}$$

where $y_s$ is the labels to $x_s$; $\mathcal{L}(f(x_s, \tilde{\theta}), y_s)$ is the one-shot loss; $\mathcal{L}(\cdot, \cdot)$ is the loss function for the deepfake detector, i.e., the AM-Softmax loss [52] in this paper inspired by the previous work [35];

---

**Algorithm 2** Offline meta-training

---

**Require:** Meta-batch size $T$, learning rates $\lambda$ and $\gamma$, training dataset $\mathbf{D}$.

1: **while** not done **do**
2:     **for** each $t$ in $T$ **do**
3:         Sample $x_r^1$ and $x_r^2$ from $\mathbf{D}$, and synthesize forgeries $x_1^f$, $x_2^f$ by blending $x_r^1$, $x_r^2$
4:         Obtain support set $(x_s, y_s)$ and query set $(x_q, y_q)$ using our split strategy
5:         Evaluate the inner loss $L_t(f(x_s, \theta), y_s)$
6:         Compute adapted parameters with gradient descent: $\theta_t' = \theta - \gamma \nabla_\theta L_t(f(x_s, \theta), y_s)$
7:     **end for**
8:     Evaluate the meta loss $\sum_t L_t(f(x_q, \theta_t'), y_q)$
9:     Update the models with gradient descent: $\theta \leftarrow \theta - \lambda \nabla_\theta \frac{1}{T} \sum_t L_t(f(x_q, \theta_t'), y_q)$
10: **end while**

**Return:** $f(\cdot, \tilde{\theta})$

---

$\gamma$ is the step size. After the updating step, we can obtain the final result by applying the updated detector $f(\cdot, \theta^*)$ to the test sample $x_e$. The algorithm of the OST scheme is shown in Algorithm 1.

**Understand the effect of one-shot test-time training:** The proposed test-time-training algorithm can also be understood as a domain adaptation method. Specifically, in OST, each image is viewed as a specific domain defined by its content, and the unseen test image will have a domain gap to the training data. The pseudo-training sample created by using the above procedure is more relevant to the test image than the training samples since $x_o^f$ is synthesized based on the test image. Thus a fast training on the sample could make the detector adapt better to the test image. More evidence on the above analysis is presented in Sec. 5.2.

### 3.2 Offline Meta-training

The proposed offline meta-training step is to ensure that a good initialization can be learned for the online OST step. To this end, a MAML [19] style meta-training scheme is utilized. Specifically, we construct a training episode by first randomly sampling two images $x_r^1$ and $x_r^2$ from $\mathbf{D}$, and then create two synthesized forgery images $x_1^f$ and $x_2^f$ with different blending operations, say, one with alpha blending and one with Poisson blending. Then we will use $x_r^1$ and $x_1^f$ to perform one-shot training. The updated model is evaluated on $x_r^2$ and $x_2^f$, and the incurred loss is used to update the model parameters with step size $\lambda$. In the above design, $x_r^1$ and $x_1^f$ are akin to the support set, and $x_2^f$ and $x_r^2$ are akin to the query set in standard meta-learning. Note that we include both $x_r^2$ and $x_2^f$ in the query set. They represent two different scenarios: $x_r^2$ is an image from the original training domain while $x_2^f$ represents an image from a new (synthesized randomly) domain. Including both the two images encourages the model works well for the existing domain and be able to generalize to a new domain. Formally, the updating process in one episode can be written as:

$$\theta \leftarrow \theta - \lambda \sum_{x_q \in \{x_r^2, x_2^f\}} \nabla_\theta \mathcal{L}(f(x_q, \theta'), y_q), \quad s.t. \quad \theta' = \theta - \gamma \sum_{x_s \in \{x_r^1, x_1^f\}} \nabla_\theta \mathcal{L}(f(x_s, \theta), y_s). \quad (2)$$

Eq. 2 is also referred to as meta update and inner update in a standard MAML-like paradigm, where $\mathcal{L}(f(x_q, \theta'), y_q)$ and $\mathcal{L}(f(x_s, \theta), y_s)$ are referred to as the meta and inner losses. The overall algorithm for the offline meta-training is illustrated in Algorithm 2.

## 4 Experiments

This section first presents the setups and then shows extensive experimental results to demonstrate the superiority of our approach. Please refer to the supplementary material for more experimental results.

### 4.1 Settings

**Training and test datasets.** Following the protocols in existing deepfake detection methods [28, 35, 57], we use the data in the Faceforencis++ (FF++) dataset [43] for training. This dataset contains

| Method | DF | | | F2F | | | FS | | | NT | | | Avg. |
|---|---|---|---|---|---|---|---|---|---|---|---|---|---|
| | DFDC | DFD | DF1.0 | DFDC | DFD | DF1.0 | DFDC | DFD | DF1.0 | DFDC | DFD | DF1.0 | |
| Xception [43] | 0.654 | 0.808 | 0.617 | 0.708 | 0.695 | 0.745 | 0.708 | 0.657 | 0.605 | 0.646 | 0.724 | 0.838 | 0.700 |
| Face X-ray [28] | 0.609 | 0.811 | 0.668 | 0.633 | 0.679 | 0.766 | 0.646 | 0.625 | 0.795 | 0.613 | 0.645 | 0.866 | 0.696 |
| F3Net [42] | 0.682 | 0.812 | 0.658 | 0.679 | 0.729 | 0.761 | 0.679 | 0.641 | 0.651 | 0.672 | 0.826 | 0.932 | 0.727 |
| RFM [50] | **0.758** | 0.833 | 0.717 | 0.736 | 0.711 | 0.732 | 0.714 | 0.593 | 0.714 | 0.726 | 0.816 | 0.846 | 0.741 |
| SRM [35] | 0.679 | 0.855 | 0.720 | 0.687 | 0.801 | 0.775 | 0.671 | 0.705 | 0.771 | 0.656 | 0.791 | **0.936** | 0.754 |
| Ours | 0.757 | **0.869** | **0.938** | **0.798** | **0.880** | **0.947** | **0.802** | **0.824** | **0.909** | **0.752** | **0.841** | 0.929 | **0.854** |

Table 1: Generalizability comparisons with state-of-the-art methods in the term of AUC. The best results are in bold. The first row denotes the training data, and the second row shows the corresponding test dataset. Our method performs favorably among the models compared.

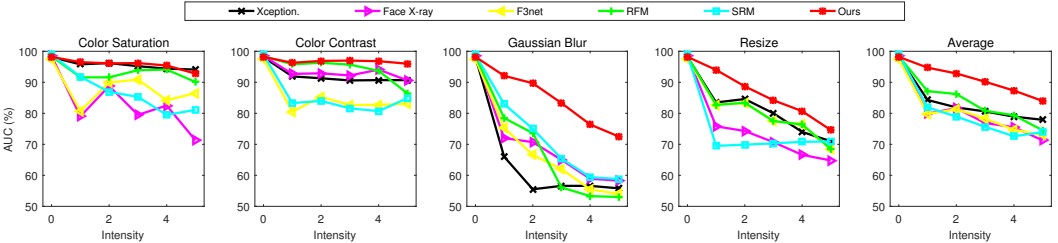

Figure 3: Generalizability comparisons under different post-processing operations in the term of AUC. The compared methods undergo five different levels of four particular types of post-processing steps. "Average" denotes the mean across all post-processing steps at each intensity level. Our method performs competitively against existing arts.

1000 videos, in which 720 videos are used for training, 140 videos are reserved for verification, and the rest are used for test. Each real video in this dataset are manipulated with four deepfake methods, including DeepFake (DF) [13], Face2Face (F2F) [49], FaceSwap (FS) [18], and NeuralTexture (NT) [48], to generate four fake videos. Each video in FF++ is processed to have three video qualities, namely raw, lightly compressed (HQ), and heavily compressed (LQ), and the HQ version is adopted by default unless otherwise stated.

To comprehensively evaluate the generalizability of our method, we use four more benchmark datasets for test, including the DeepfakeDetection (DFD) [12] which consists of 363 real and 3068 fake videos created by the improved DF technique; Deepfake Detection Challenge (DFDC) [15] that contains over 1,000 real and over 4,000 fake videos manipulated by multiple Deepfake, GANbased, and non-learned methods; DeeperForensics-1.0 (DF1.0) [25] that is composed of over 11,000 fake videos generated by their DFVAE method, and the CelebDF [32] dataset that contains 408 real videos and 795 synthesized videos generated by the improved DF method. The forgeries in the training and test datasets do not overlap both in their contents and creating methods.

**Implementation details.** We use Xception [11] as the backbone network, and the parameters are initialized by the weights pretrained on the ImageNet [14]. We use DLIB [44] for face extraction and alignment, and we resize the aligned faces to $256 \times 256$ for all the sampled frames in the training and test datasets. We use the Adam optimizer [26] for optimizing the network with $\beta_1 = 0.9$ and $\beta_1 = 0.999$, and the meta-batch size is set to be 20. The learning rates for the inner update (*i.e.* $\gamma$ in Eq. (1) and (2)) and meta update (*i.e.* $\lambda$ in Eq. (2)) are fixed as 0.0005 and 0.0002 for both the offline and online training phases.

## 4.2 Generalizability Comparisons

We compare our method with several state-of-the-arts, including Xception [43], Face X-ray [28], F3net [42], RFM [50], and SRM [35], to demonstrate its superior generalizability. To ensure fair comparisons, we use the provided codes of Xception, RFM, and SRM from the authors, and we reimplement Face X-ray and F3Net rigorously following the companion paper's instructions and train these models under the same settings.

| Method | Training dataset | DFDC | | | CelebDF | | |
|--------|------------------|------|------|------|---------|------|------|
| | | AUC ↑ | ACC ↑ | ERR ↓ | AUC ↑ | ACC ↑ | ERR ↓ |
| MLDG [27] | | 0.682 | 0.607 | 0.370 | 0.609 | 0.595 | 0.418 |
| LTW [45] | FF++ | 0.690 | 0.631 | 0.368 | 0.641 | 0.634 | 0.397 |
| MT3 [5] | | 0.775 | 0.667 | 0.307 | 0.701 | 0.664 | 0.319 |
| Ours | | **0.833** | **0.714** | **0.250** | **0.748** | **0.673** | **0.312** |

Table 2: Comparisons with models based on meta learning in terms of AUC, ACC, and ERR.

| | [59] | [1] | [31] | [37] | [54] | [38] | [29] | [36] | [33] | [56] | Ours |
|--|------|-----|------|------|------|------|------|------|------|------|------|
| FF++ | 0.701 | 0.847 | 0.930 | 0.664 | 0.473 | 0.966 | 0.968 | 0.932 | 0.969 | **0.998** | 0.982 |
| CelebDF | 0.538 | 0.548 | 0.646 | 0.550 | 0.546 | 0.575 | 0.563 | 0.734 | 0.724 | 0.674 | **0.748** |

Table 3: Extensive evaluations with other state-of-the-arts (*i.e.* Two-stream [59], MesoNet [1], FWA [31], VA-MLP [37], Headpose [54], Capsule [38], SMIL [29], Two-branch [36], SPSL [33], and MADD [56]) in the term of AUC. The compared models are trained on the FF++ dataset [43] while testing on both FF++ [43] and CelebDF [32]. Our method performs favorably against existing arts.

**Generalizing to different datasets.** Forgeries in different datasets are often created by different deepfake methods. In a real-world scenario, suspicious images are likely to be created by unseen methods from unseen sources, thus the generalizability to different datasets would be crucial. To evaluate the generalizability, we conduct experiments by training the compared methods on each of the four data in FF++ (*i.e.* DF, F2F, FS, and NT) and test them on the benchmark datasets, including DF1.0, DFD, DFDC. The experimental setting is not trivial since all the deepfake methods that synthesize the test forgeries are unseen in the training dataset.

Experimental results in the term of Area Under Curve (AUC) are listed in Table 1. Existing arts [43, 28, 42, 35] rely on imperceptible artifacts for generalizing, but these artifacts may appear with different patterns in forgeries from different datasets, thus limiting their generalizing. RFM [50] suggests mining more regions in the detected images for classifying. However, this approach still cannot avoid the domain gap between the training and test samples. In comparison, our method suggests finetuning the pretrained detector on the test data, and it enables adaptation to the new data before evaluation, thus improving generalizing. Meanwhile, we can observe that our method outperforms other models in most cases, and it surpasses the second best [35] by $10\%$ in the term of overall performance, which clearly demonstrates the advantages of the proposed OST strategy for generalizing to new forgeries.

**Generalizing to different post-processing steps.** For real-world deepfake detection requirements, it is of paramount importance that a detector can generalize to different post-processing steps. We conduct experiments with different post-process steps to evaluate the generalizability of our method. Specifically, we train the compared models on the original FF++ data and test them on FF++ samples that were exposed to various unseen post-processing steps.

Following [25], we consider four popular post-processing steps, including changes in color saturation, changes in color contrast, Gaussian blur, and resize: downsample the image by a factor then upsample it to the original resolution. Note that all the models are trained without these augmentations, so that these post-processing steps are unseen during training. The results are plotted in Figure 3. It is evident that our method can better generalize to unseen post-process operations than other models [43, 28, 42, 35, 50]. In particular, the changes in color saturation and color contrast can hardly affect our model, while state-of-the-arts suffer from performance declines to different extents. Moreover, OST maintains higher performances than all the compared models when the post-processing steps affect the high frequency content of the images (*i.e.* blurring and resizing), where existing arts undergo significant deterioration. We attribute this to their reliance on the imperceptible low-level clues, which will be damaged by these steps. We also include comparisons regarding another post-processing step (*i.e.* compression), please refer to the supplementary material for details.

### 4.3 State-of-the-art Comparisons

**Comparisons with other models based on meta learning.** The idea of using meta learning to boost the generalization of the deepfake detector has been explored in the previous work, such as that in

| Method | DF | | | F2F | | | FS | | | NT | | | Avg. |
|---|---|---|---|---|---|---|---|---|---|---|---|---|---|
| | DFDC | DFD | DF1.0 | DFDC | DFD | DF1.0 | DFDC | DFD | DF1.0 | DFDC | DFD | DF1.0 | |
| Baseline | 0.742 | 0.841 | 0.910 | 0.693 | 0.816 | 0.924 | 0.735 | 0.750 | **0.913** | 0.736 | **0.848** | 0.895 | 0.817 |
| TTT [46] | 0.717 | 0.862 | 0.859 | 0.666 | 0.820 | 0.891 | 0.733 | 0.684 | 0.833 | 0.677 | 0.825 | 0.863 | 0.786 |
| TENT [51] | 0.749 | 0.851 | 0.915 | 0.707 | 0.826 | 0.916 | 0.725 | 0.752 | **0.915** | 0.748 | 0.840 | 0.886 | 0.819 |
| Ours | **0.757** | **0.869** | **0.938** | **0.798** | **0.880** | **0.947** | **0.802** | **0.824** | 0.909 | **0.752** | 0.841 | **0.929** | **0.854** |

Table 4: Ablation studies regarding the effectiveness of the proposed OST. The metric is AUC. The baseline model is our method without the online training step (but using synthesized images at the training stage). We reimplement TTT [46] and TENT [51] with our MAML-based framework using the same self-supervised task from [46] and entropy minimization task from [51].

LTW [45]. Inspired by MLDG [27], which uses meta-learning to solve the domain generalization problem, LTW assumes forgeries from different deepfake methods as different domains, so the generalizable deepfake detection task can be regarded as a domain generalizing problem. In this experiment, we compare our method with LTW and MLDG. Note that both of these models require the training dataset to contain more than one forgery creating methods. We thus train the models on the FF++ dataset and test them on CelebDF and DFDC datasets. Moreover, we also reimplement and evluate a recent meta learning-based method MT3 [5] which uses a contrastive loss between the original sample and its augmentation to update the parameters during inference.

Results are shown in Table 2, where the figures are directly cited from the reported statistics in [45]. We observe that the proposed method outperforms others in terms of ACC, AUC, and EER. The main reason is that although LTW and MLDG can learn good initializations via meta learning, their detectors cannot finetune on the unknown test domains, thus limiting the overall performances. Meanwhile, the contrastive loss may be less effective in the deepfake detection task which limits the generalizability of MT3. In comparison, our pretrained detector can adapt to the test samples via OST, and our designed pseudo training task is specially designed for the deepfake detection task, thus being more effective than other arts.

**Comparisons with other state-of-the-art detectors.** We further evaluate our method against several more state-of-the-art algorithms, including the Two-stream [59], MesoNet [1], Headpose [54], FWA [31], VA-MLP [37], Capsule [38], SMIL [29], Two-branch [36], SPSL [33], and MADD [56]. All the models are trained on the FF++ dataset and tested on FF++ and CelebDF datasets. We cite the figures from [33] directly for some compared models. Results in the term of AUC are listed in Table 3. Our method achieves favorable performance when generalizing to CelebDF while still maintaining a good result on FF++ compared with all the other methods.

# 5 Algorithm Analysis

In this section, we first conduct ablation studies to validate the effectiveness of our OST algorithm. Then, we illustrate why the pseudo samples can lead to better generalization. Besides, we discuss how sample selection affects our OST algorithm and the adaption of our method to a regular framework.

## 5.1 Ablation Studies

Our ablation studies on evaluating the proposed OST method consist of the comparisons of our model to the following two configurations. The first one is the baseline configuration that does not perform online training but include all the other operations, e.g., augmentation of domains via synthesized images. The second one is a self-supervised test-time-training (i.e., TTT algorithm in [46]). In TTT, we use MAML as offline meta-learning and a rotation prediction task [21] for training and test, which is the same as that in [46]. The same framework is applied for the third variant TENT [51] which aims to minimize the prediction entropy via inference. The data collection, split, and other implementation configurations remain the same to ensure fair comparisons.

We train the baseline method, TTT, TENT, and our method on four types of data in the FF++ dataset. Then, we evaluate these learned detector models on the DFDC, DFD, and DF1.0 datasets. Table 4 shows the evaluation results by using the AUC metric. These results indicate that our OST method improves baseline by 4%. This improvement shows the effectiveness of our online

|  | DFDC | DFD | DF1.0 | CelebDF |
|---|---|---|---|---|
| $\|f(x_r,\tilde{\theta}))-f(x_e,\tilde{\theta}))\|$ | 2.06 | 1.06 | 0.94 | 1.43 |
| $\|f(x_o^f,\tilde{\theta}))-f(x_e,\tilde{\theta}))\|$ | 1.63 | 0.82 | 0.61 | 1.03 |

Table 5: Average feature distances of the test samples (*i.e.* $x_e$) from different test sets to the training sample (*i.e.* $x_r$) and their corresponding pseudo-training forgeries (*i.e.* $x_o^f$). Distance between $x_e$ and $x_o^f$ is closer than that between $x_e$ and $x_r$ in all cases.

| Method | DF | | | F2F | | | FS | | | NT | | | Avg. |
|---|---|---|---|---|---|---|---|---|---|---|---|---|---|
| | DFDC | DFD | DF1.0 | DFDC | DFD | DF1.0 | DFDC | DFD | DF1.0 | DFDC | DFD | DF1.0 | |
| NN | 0.790 | 0.857 | 0.944 | 0.806 | 0.841 | 0.938 | 0.791 | 0.831 | 0.914 | 0.760 | 0.858 | 0.931 | 0.855 |
| Avg | 0.773 | 0.898 | 0.956 | 0.768 | 0.820 | 0.929 | 0.773 | 0.809 | 0.924 | 0.751 | 0.847 | 0.940 | 0.849 |
| X w/o OST | 0.761 | 0.836 | 0.928 | 0.729 | 0.774 | 0.912 | 0.755 | 0.784 | 0.894 | 0.739 | 0.825 | 0.906 | 0.820 |
| X w/ OST | 0.763 | 0.845 | 0.962 | 0.733 | 0.785 | 0.943 | 0.766 | 0.812 | 0.946 | 0.775 | 0.889 | 0.929 | 0.846 |
| Ours | 0.757 | 0.869 | 0.938 | 0.798 | 0.880 | 0.947 | 0.802 | 0.824 | 0.909 | 0.752 | 0.841 | 0.929 | 0.854 |

Table 6: Effects of different sampling and learning process. We select the training sample to be blended by using nearest neighbor sampling, and average sampling. The detection results are shown on the first and second rows, respectively. We use the Xception network without and with our OST scheme (*i.e.* X w/o OST and X w/ OST) to show if our method can still thrive in a regular framework. The results are shown on the third and fourth rows, respectively.

test-time training method. On the other hand, we observe that TTT does not improve the baseline performance. This corresponds to the observation in [34] that an inappropriate self-supervised task (i.e., rotation prediction task) in the TTT scheme may jeopardize the main deepfake detection performance. Meanwhile, the small entropy regularization in TENT may not be as effective in the binary classification task as it in the multi class classification task, explaining why its improvements over the baseline model is subtle. Overall, the experiments show that our OST method is more generalizable than the existing TTT-based methods [46, 51] for the deepfake detection task.

### 5.2 Explanation for OST: Pseudo Samples Can Mitigate the Diverged Data Distributions

In section 3.1, we explain that the effect of OST is by training the detector on a synthesized pseudo-training sample that is closer to the test images than the original training data. This is due to the synthesized image being created based on the test image. In this subsection, we further provide some experimental evidence for this argument.

We experimentally validate the above analysis by computing the feature distance of test samples (i.e., $x_e$) to the training samples (*i.e.* $x_r$), and their corresponding pseudo-training samples (*i.e.* $x_o^f$). We use test samples from the four test sets (*i.e.* DFDC, DFD, DF1.0, CelebDF) for evaluation, and the euclidean distance is adopted as the evaluation metric. Table 5 show the results. We observe that in all test sets, the average distance between $x_o^f$ and $x_e$ is smaller than that between $x_r$ and $x_e$. These results consistently correspond to our aforementioned analysis. As a result, we can conclude that the generated pseudo-training samples are with features that reside in the intermediate state of the training and test data. By finetuning the detector with pseudo-training samples that are close to test samples, we can make the model better adapt to the domain defined by the test sample.

### 5.3 Further Analysis

**Sample selections.** During forgery synthesis, we randomly select one training sample to blend with the test sample. To investigate whether the random selection is an effective approach, we conduct ablation studies where our random selection is compared to the other two configurations. The first configuration is the nearest neighbor sampling (*i.e.* NN sampling). For each test sample, we select the training sample whose feature distance is closest for blending. The second configuration is the average sampling (*i.e.* Avg sampling), where we average the training sample performance for evaluation. We use these configurations on four types of data on FF++ datasets, and evaluate their performance on DFDC, DFD, and DF1.0 datasets. The first and second rows of Table 6 show the results. We observe that the differences of the average performance among these three sampling

| Steps | DF | | | F2F | | | FS | | | NT | | | Avg. | TC(s) |
|---|---|---|---|---|---|---|---|---|---|---|---|---|---|---|
| | DFDC | DFD | DF1.0 | DFDC | DFD | DF1.0 | DFDC | DFD | DF1.0 | DFDC | DFD | DF1.0 | | |
| 1 step | 0.757 | 0.869 | 0.938 | 0.798 | 0.880 | 0.947 | 0.802 | 0.824 | 0.909 | 0.752 | 0.841 | 0.929 | 0.854 | 0.065 |
| 2 step | 0.759 | 0.902 | 0.927 | 0.794 | 0.892 | 0.942 | 0.797 | 0.830 | 0.922 | 0.758 | 0.836 | 0.947 | 0.859 | 0.121 |
| 3 step | 0.783 | 0.860 | 0.942 | 0.804 | 0.906 | 0.934 | 0.801 | 0.840 | 0.929 | 0.747 | 0.843 | 0.948 | 0.861 | 0.181 |

Table 7: Performances of using different gradient descent steps. Improvements brought by more updating steps are subtle, while the time consumptions (TC) are proportional to the updating steps.

strategies (*i.e.* NN sampling, Avg sampling, and random sampling) are small (less than 0.01). The small difference indicates that the sampling strategies during inference are not critical in our model. To this end, we use the random sampling strategy for efficiency considerations.

**OST adaptions.** Our learning framework is based on MAML which can adapt to new tasks efficiently. We analyze how our OST method functions when we substitute MAML with a regular training scheme. The training data is kept the same, and the backbone is still Xception. Results are shown in the third and fourth rows of Table 6. We observe that OST with regular end-to-end network training performs on par with the MAML framework. On the other hand, the detection performance decreases when the model does not employ our OST process. These results indicate that our method suites regular training configurations and improves prevalent network backbones. However, the MAML framework enables a faster adaption during online inference (i.e., 10 times faster for each test sample). In practice, we utilize the MAML framework and integrate our OST method.

**Multiple Gradient Descent Steps.** To study if more updating steps can improve the detection, we conduct ablation studies by using different gradient descent steps during evaluation. The model is trained on the FF++ dataset, and it is evaluated on the DFDC, DFD, and DF1.0 datasets. As shown in Table 7, the accuracies of using more gradient descent steps are not improved greatly while the computational times are proportional to the gradient descent steps. Meanwhile, using more gradient descent steps also requires much more memory costs in an MAML-based framework. We thus use only one gradient descent step in our method as a compromise between efficiency and accuracy.

## 6  Conclusion and Discussions

In this paper, we propose a new learning paradigm specially designed for the generalizable deepfake detection task. To recap briefly, for each test sample, we suggest finetuning the pretrained detector with a pseudo-training sample, which is synthesized by blending the test samples and a randomly selected template image, before the classification step. We empirically find that the proposed online training scheme enables the pretrained model to adapt to the sample-specific statistics, thus facilitating the generalizability. We implement our method in a MAML-based scheme to enable fast adaptation to different test samples, which shows superior performance against state-of-the-art algorithms for generalization to unseen forgeries and various post-processing operations.

**Limitations and future works.** As the pseudo-training samples are synthesized following existing deepfake pipelines, our method cannot be applied to the scenario when the fake images are created following other protocols, such as the case when fake images are totally synthesized by GAN-based methods. It is our future work to develop methods that aim for both deepfakes and GAN-synthesized fake images. Meanwhile, DLIB is used in our forgery synthesizing pipeline to detect and extract facial landmarks. Considering that there are circumstances that DLIB may fail, which may also fail OST at the same time. Thus, a more effective facial detecting method is promotive for OST.

**Ethic statement.** This work is designed to help people better fight against the abuse of deepfake technology. It does not involve any human or animal subjects, and there is no violation of personal privacy while conducting experiments. We do not anticipate any potentially harmful consequences to our work. Through our study and releasing our code, we hope to raise stronger research and societal awareness towards the problem of generalizable deepfake detection.

**Acknowledgements.** Liang Chen is supported by the China Scholarship Council (CSC Student ID 202008440331).

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
