# Supplementary Material for
# OST: Improving Generalization of DeepFake Detection via One-Shot Test-Time Training

**Liang Chen**[1]    **Yong Zhang**[2*]    **Yibing Song**[2]    **Jue Wang**[2]    **Lingqiao Liu**[1*]
[1] The University of Adelaide    [2] Tencent AI Lab
{liangchen527, zhangyong201303, yibingsong.cv, arphid}@gmail.com
lingqiao.liu@adelaide.edu.au

In this supplementary material, we provide,

1. Preprocessing steps and details in Section 1.

2. Additional experiments regarding the generalizability comparisons under different image compression levels and resolutions in Section 2;

3. Evaluations with calibration and other metrics in Section 3;

4. Extending of our method to the video detection task in Section 4.

5. Time consumption and computational complexity comparisons against other models in Section 5.

6. Visualizations of the embedded representations in Section 6.

## 1   Preprocessing steps and Details

Following the settings in FF++ [8], we extract 270 frames from each training video to construct the training set. The extracted and aligned face images are resized to the size of $256 \times 256$ and are then normalized to the range of $[-1, 1]$. For our model and all the reimplement methods, we do not apply any data augmentation steps for the training data. We implement the model in PyTorch, and we compute the AM-Softmax loss [11] with cosine margin under the default hyper-parameter settings ($\gamma = 0.0$, $m = 0.5$, $s = 30$, and $t = 1.0$). All the methods are running on the same machine (a Nvidia Tesla v100 GPU with 32G memory).

## 2   Generalizations to Different Compression Levels and Resolutions

**Generalizations to different compression levels.** Given the fact that real-world images are usually stored by different compression levels, it is crucial that deployed deepfake detectors are not easily subverted by the varying image compression levels. To evaluate the generalizability of the compared models [8, 5, 7, 6] regarding different compression levels, we separately train them on the F2F data and test them on the DF and FS data with different compression levels.

The results in terms of ACC and AUC are listed in Table 1. We note that when the training and test images are with the same compression level, almost all the compared models can obtain remarkable performances. However, some methods [8, 5, 6] suffer from large performance drop when trained on the HQ data while test on the LQ data. The results are not surprising. The main reason is that these methods rely on the low-level image patterns which will be largely destroyed when the images are highly compressed. On the contrary, the proposed method is substantially less affected by the compression levels, outperforming all other methods. The results demonstrate the effectiveness of our method when confronting different image compression levels.

---

*Corresponding authors. This work is done when L. Chen is an intern in Tencent AI Lab.

36th Conference on Neural Information Processing Systems (NeurIPS 2022).

| Training set | Method | Test set | | | |
|---|---|---|---|---|---|
| | | LQ | | HQ | |
| | | DF | FS | DF | FS |
| F2F (LQ) | Xception | 0.621 / 0.666 | 0.557 / 0.504 | 0.666 / 0.698 | 0.536 / 0.559 |
| | Face X-ray | 0.570 / 0.675 | 0.549 / 0.511 | 0.589 / 0.690 | 0.517 / 0.529 |
| | F3Net | 0.631 / 0.698 | 0.554 / 0.560 | 0.629 / 0.719 | 0.546 / 0.578 |
| | RFM | 0.605 / 0.699 | 0.573 / 0.580 | 0.637 / 0.732 | 0.549 / 0.627 |
| | SRM | 0.631 / 0.727 | 0.561 / 0.552 | 0.627 / 0.753 | 0.528 / 0.550 |
| | Ours | **0.718 / 0.809** | **0.609 / 0.663** | **0.786 / 0.896** | **0.584 / 0.634** |
| F2F (HQ) | Xception | 0.544 / 0.546 | 0.516 / 0.511 | 0.648 / 0.749 | 0.695 / 0.753 |
| | Face X-ray | 0.518 / 0.578 | 0.529 / 0.525 | 0.558 / 0.662 | **0.813 / 0.859** |
| | F3Net | 0.529 / 0.571 | 0.517 / 0.521 | 0.663 / 0.798 | 0.565 / 0.677 |
| | RFM | 0.529 / 0.556 | 0.518 / 0.515 | 0.604 / 0.794 | 0.530 / 0.646 |
| | SRM | 0.576 / 0.592 | 0.536 / 0.533 | 0.574 / 0.792 | 0.695 / 0.754 |
| | Ours | **0.695 / 0.750** | **0.591 / 0.606** | **0.853 / 0.950** | 0.754 / 0.826 |

Table 1: Additional generalizability comparisons across different compression levels in terms of ACC / AUC. Our method achieves comparable performance against existing methods. LQ and HQ denote the highly compressed and lightly compressed data in the FF++ datasets [8].

| Resolution | DF | | | F2F | | | FS | | | NT | | | Avg. |
|---|---|---|---|---|---|---|---|---|---|---|---|---|---|
| | DFDC | DFD | DF1.0 | DFDC | DFD | DF1.0 | DFDC | DFD | DF1.0 | DFDC | DFD | DF1.0 | |
| $200 \times 200$ | 0.741 | 0.855 | 0.962 | 0.782 | 0.862 | 0.959 | 0.790 | 0.794 | 0.947 | 0.713 | 0.812 | 0.937 | 0.846 |
| $320 \times 320$ | 0.755 | 0.916 | 0.937 | 0.721 | 0.858 | 0.948 | 0.843 | 0.801 | 0.939 | 0.760 | 0.823 | 0.931 | 0.853 |
| $256 \times 256$ | 0.757 | 0.869 | 0.938 | 0.798 | 0.880 | 0.947 | 0.802 | 0.824 | 0.909 | 0.752 | 0.841 | 0.929 | 0.854 |

Table 2: Ablation studies regarding using images with different resolutions.

**Evaluations using images with different resolutions.** Following the setting in previous works [5, 6], faces from the training and test datasets are resized to $256 \times 256$. To evaluate if the resolution can also influence the generalizability of the detector, we conduct ablation studies by using samples with different resolutions during training and test. Specifically, we evaluate our model on images with resolutions of $200 \times 200$ and $320 \times 320$. Results are listed in Table 2. We observe that the differences between the three different resolutions are rather subtle, indicating that image resolution is not an influential factor for generalization.

## 3 Evaluations of Different Methods with Calibration and Other Metrics

**Calibration for better generalizing.** Following the setting in [12], we calibrate the model with a randomly selected pristine and deepfake pair from the test dataset. We first augment the image pair for 128 times to obtain a small calibration set. Then the small calibration set is passed into our detection model to get the logits, which are then fitted by a logistic regression. Similarly, we take the weight and bias learned from the logistic regression to adjust the output of our model. After applying the bias and a sigmoid function to the output logit, we can finally obtain the calibrated probability. Comparisons of accuracies from different methods before and after calibration are listed in Table 4. We observe that our method still outperforms existing methods for both calibrated and uncalibrated results. Meanwhile, we note the calibration does not improve the accuracy in all cases. This phenomenon is consistent with the observations in [12].

**Evaluation with other metrics.** To evaluate how the compared methods perform on pristine and deepfake separately, we also report true positive (TP), true negative (TN), false negative (FN), false positive (FP), and true negative rate (TNR) for them. The methods are trained with the FF++ dataset and evaluated using DFDC, DFD, and DF1.0 datasets. Results are shown in Table 3. We observe that the TNR of our method is much larger than other methods, indicating that our method is more likely to detect a given deepfake image. In comparison, TNRs for other methods are around 0.5, which shows that these methods can hardly identify a fake image.

| Dataset (P/N) | DFDC (1533/1613) | | | | DFD (921/7963) | | | | DF1.0 (10050/10050) | | | | Avg. |
|---|---|---|---|---|---|---|---|---|---|---|---|---|---|
| Metric | TP | TN | FN | FP | TP | TN | FN | FP | TP | TN | FN | FP | TNR |
| Xception [8] | 1410 | 700 | 123 | 913 | 783 | 5375 | 138 | 2588 | 8857 | 2064 | 1193 | 7986 | 0.415 |
| Face X-ray [5] | 1288 | 784 | 245 | 829 | 814 | 4984 | 107 | 2979 | 9435 | 3118 | 615 | 6932 | 0.453 |
| F3Net [7] | 1387 | 679 | 146 | 934 | 902 | 4465 | 19 | 3498 | 9338 | 4880 | 712 | 5170 | 0.511 |
| RFM [10] | 1109 | 1192 | 424 | 421 | 763 | 6191 | 158 | 1772 | 9731 | 2835 | 319 | 7215 | 0.521 |
| SRM [6] | 1348 | 841 | 185 | 772 | 874 | 5302 | 47 | 2661 | 10037 | 3712 | 13 | 6338 | 0.502 |
| Ours | 1032 | 1214 | 501 | 399 | 781 | 6602 | 140 | 1361 | 8336 | 9816 | 1714 | 234 | 0.898 |

Table 3: Performances on pristine and deepfake separately. Here TP, TN, FP, and FN represent true positive, true negative, false positive, and false negative, respectively. P/N are the numbers of the positive (pristine) and negative (deepfake) samples in the corresponding dataset.

| Method | DFDC | | DFD | | DF1.0 | |
|---|---|---|---|---|---|---|
| | Original | Calibrated | Original | Calibrated | Original | Calibrated |
| Xception [8] | 0.671 | 0.704 | 0.693 | 0.688 | 0.543 | 0.622 |
| Face X-ray [5] | 0.659 | 0.674 | 0.653 | 0.649 | 0.625 | 0.684 |
| F3Net [7] | 0.657 | 0.691 | 0.604 | 0.659 | 0.707 | 0.693 |
| RFM [10] | 0.731 | 0.702 | 0.783 | 0.796 | 0.625 | 0.649 |
| SRM [6] | 0.696 | 0.688 | 0.695 | 0.714 | 0.684 | 0.677 |
| Ours | 0.714 | 0.703 | 0.831 | 0.858 | 0.903 | 0.881 |

Table 4: Performances comparisons for detectors before and after calibration. We show the accuracy of recent detectors trained on the FF++ dataset and test them on different benchmarks. The calibration refers to the two-shot classifier calibration in [12], and the original results are uncalibrated.

## 4 Extending to Video Detection

Recent studies have also designed deepfake detectors especially for the video detection task. Different from the image detector which takes a single image as the input, the video detector usually takes a sequence of video frames as input. Moreover, besides the spatial information, temporal information hidden in the consecutive frames is also mined for the video detectors. But it is noteworthy that any image detectors can be extended to the video detection task theoretically, while the video detectors cannot be used for the image classification task.

In this section, we extend our image detector for the video detection task. Specifically, for each video, we sample one frame every five frames to collect a total of twenty five images. Then, we average the predictions over sampled images for the final classification of each video. Following the protocol in [4], we test the compared models on each of the four methods in the FF++ dataset after training on

| | Method | Training on remaining three | | | | |
|---|---|---|---|---|---|---|
| | | DF | F2F | FS | NT | Avg. |
| video detector | CNN-GRU | 0.976 | 0.858 | 0.476 | 0.866 | 0.794 |
| | LipForensics scratch | 0.930 | 0.988 | 0.567 | 0.983 | 0.867 |
| | LipForensics finetune | 0.984 | 0.994 | 0.804 | **0.993** | 0.944 |
| | LipForensics | **0.997** | **0.997** | **0.901** | 0.991 | **0.971** |
| image detector | Xception | 0.939 | 0.868 | 0.512 | 0.797 | 0.779 |
| | CNN-aug | 0.875 | 0.801 | 0.563 | 0.678 | 0.729 |
| | Patch-based | 0.940 | 0.873 | 0.605 | **0.848** | 0.817 |
| | Ours | **0.991** | **0.942** | **0.899** | 0.826 | **0.915** |

Table 5: Generalization comparisons in the term of AUC with state-of-the-art methods when extended to the video detection task. The image detectors use average results from different frames within a video for classification. The compared methods are tested on each forgery data in the FF++ dataset while training on the remaining three. The results of CNN-GRU [9], LipForencis [4], Xception [8], CNN-aug [12], and Patch-based [1] are directly cited from [4].

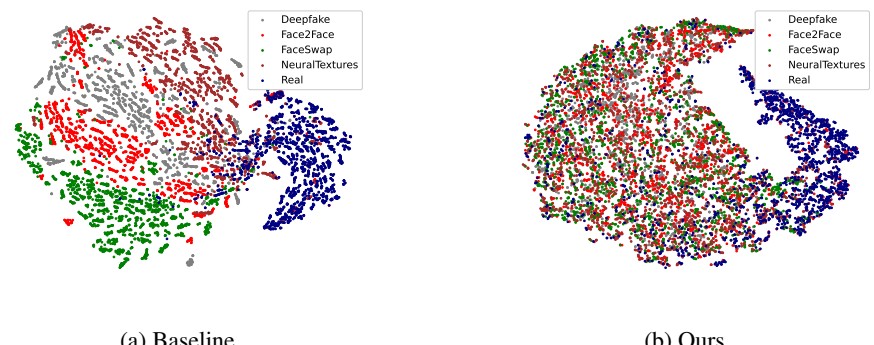

|  | (a) Baseline |  | (b) Ours |  |

Figure 1: T-SNE visualization of features from different models.

|  | Xceptiion [8] | Face X-ray [5] | F3Net [7] | RFM [10] | SRM [6] | Ours |
|---|---|---|---|---|---|---|
| TC (s) | 0.015 | 0.017 | 0.019 | 0.015 | 0.037 | 0.062 + 0.074 |
| CC (MACs(G)) | 6.01 | 6.01 | 6.05 | 6.01 | 13.81 | 18.03 |

Table 6: Time consumption (TC) and computational complexity (CC) comparisons. All methods are evaluated on the same device with a $256 \times 256$ image. The inference time of our method takes 0.062 seconds, and our pseudo sample generating process takes 0.074 seconds. Our method takes two forwards and one backward, thus requiring more flops.

the remaining three. Table 5 lists the results from both the image detectors and video detectors. It is not surprising that the state-of-the-art video detector outperforms all the methods since the temporal information is well explored in their algorithms. Meanwhile, we note our method outperforms other image detectors by a large margin, and it also performs competitively against the video predictor (*i.e.* CNN-GRU [9] and LipForencis scratch [4]) even without any temporal information. The results demonstrate the effectiveness of our method when extended to the video detection task.

## 5   Time Consumption and Computational Complexity Comparisons

To comprehensively evaluate the proposed method, we provide the time consumption and computational complexity comparisons in Table 6. All the compared methods are evaluated on the same device using a $256 \times 256$ image. All the models are implemented with the Xception baseline [3], thus the computational complexities are nearly the same for most models except for SRM [6] which uses a dual branch network architecture. Because our model includes two forwards and one backward operations during inference, thus the corresponding computational complexity is more than others. Meanwhile, our method involves the generation of pseudo training samples which uses CPU for the task mostly (except for the learning-based generating method [2]). Thus, it requires much more running time than others. However, it is noteworthy to point out that in this relatively early stage, the deepfake detection community considers little on time consumption but more on the accuracy. We will explore possible speedup techniques such as model compression and distillation in future work.

## 6   Visualizations of the Embedded Representations

This section presents the plots of the 2D orthogonal projection of the extracted patterns from our model and that from the baseline model (i.e. Xception [8]). Specifically, we show the t-SNE visualization of the extracted features from the FF++ dataset with different detectors in Figure 1. We observe that both the two models can well separate pristine and deepfake, but the distributions of real and fake representations are rather different. Although the different forgeries are all considered one class, the baseline detector seems to reveal unique artifacts of each face manipulation algorithm as shown in Figure 1 (a). In contrast, the fake representations extracted from our OST model are more mixed together, which indicates our model tends to explore more common features for classification, which can boost generalization when encountering forgeries created by unknown methods.