# OpenReview forum: "OST: Improving Generalization of DeepFake Detection via One-Shot Test-Time Training"
_NeurIPS.cc/2022/Conference — NeurIPS 2022 Accept_

### Official Review · Reviewer_xdPp · 2022-07-08

**Rating:** 6
**Confidence:** 3
**Soundness:** 3 good
**Presentation:** 3 good
**Contribution:** 3 good

**Summary:**

This paper proposes a new method for generalized deepfake detection in which the testing has domain gap from the training. The proposed method is based on meta-learning on training set and one-shot test-time training on the synthesized pseudo samples. Extensive experiments show that the proposed method outperforms the state of the art. Ablation study proves the effectiveness of the proposed designs.

**Questions:**

Will the result differ a lot for using different x_r in OST? The authors can provide further analysis on it.

**Limitations:**

The authors have discussed the limitations. No potential negative societal impact.

**Strengths And Weaknesses:**

Strength:

1.The paper is well-written. It has good literature review which clearly shows the difference between this paper and related works.

2.The proposed method is interesting and well designed. The motivation is clear and reasonable.

3.Extensive experiments with analysis are provided. The new method outperforms a number of state-of-the-art methods.

4.The experiment details are complete.


Weaknesses:

I didn’t see obvious drawbacks.

The authors can provide comparison to more SOTA methods in table 2.

The paper can be further enhanced by illustrating the advantages brought by the proposed OST adaptation in visualized examples.

---

> ### Author Response · Authors · 2022-08-02
> **Including more methods in Table 2, illustrating visual examples and discussion of using different template sample in OST**
>
> We thank the reviewer for the comments, and we answer the raised questions below.
>
> * **1. Including more methods in Table 2**
>
> We include a comparison in Table 2 by comparing OST with a recent meta learning-based method MT3 [5] which uses a contrastive loss between the original sample and its augmentation to update the parameters during inference. The results are listed below. We observe that our method performs favorably against MT3 [5] in both benchmarks in all evaluation metrics.
>
> ||| DFDC|| | CelebDF| ||
> |--- |--- | --- | --- | --- |--- | --- | --- |
> |Method| Training dataset | AUC |ACC  |ERR|AUC |ACC  |ERR|
> |MLDG| FF++|0.682|0.607|0.370|0.609|0.595|0.418|
> |LTW | FF++|0.690|0.631|0.368|0.641|0.634|0.397|
> |MT3| FF++|0.775|0.667|0.307|0.701|0.664|0.319|
> |Ours| FF++|0.833|0.714|0.250|0.748|0.673|0.312|
>
> * **2. Visual example**
>
> We include t-SNE visualizations to demonstrate the advantages of OST over the baseline model. Please refer to Section G and Figure 4 in the appendix of the revised manuscript for a detailed description.
>
> * **3. Using different template sample $x_r$**
>
> We provide ablation studies using different template samples in Section 5.3 of our manuscript. We replace the random selection strategy with two variants. The first is using nearest neighbor sampling, which selects the template image that is closest to the test sample. Another is the average sampling strategy, where we sample five different $x_r$ and use them with the current test sample to synthesize five different pseudo training samples. We report the average accuracy from the detector finetuned with these five different pseudo training samples. Results in the first and second rows of Table 6 indicate that different template samples do not bring many differences for the model.

---

> > ### Comment · Reviewer_xdPp · 2022-08-05
> > **Thanks for the response.**
> >
> > Thanks for the response. The modification does enhance this paper. It is interesting to see the reasonable finding in feature visualization.

---

### Official Review · Reviewer_DZTC · 2022-07-09

**Rating:** 5
**Confidence:** 5
**Soundness:** 3 good
**Presentation:** 3 good
**Contribution:** 3 good

**Summary:**

This paper studies a Test-Time Training paradigm to improve the generalization ability of deepfake detection method to unseen forgery attacks. Specifically, for each coming test sample, the proposed method synthesizes a pseudo-training sample by blending the test samples with a randomly selected template image and finetunes the pretrained detector with it. To achieve a good initialization for test time training, this paper adopts the MAML-based meta-training to enable fast adaptation to different test samples. Experiments are carried out in several public deepfake datasets.

**Questions:**

1. In line 136, how many x_r are sampled? Could the authors provide more details about the number or proportion of template images and the synthesized images in the One-shot online training?

2. What is the technical novelty of proposed Offline Meta-training?

3. How long does Online Test-Time Training take?

4. Why only perform a single-step gradient descent? How is the performance if the number of step gradient descent increases?


**Limitations:**

Yes.

**Strengths And Weaknesses:**

Strengths:

1. Studying Test-time Training to improve the generalization ability of deepfake detection is reasonable.

2. Generating pseudo-training samples in the proposed Online Test-Time Training is novel and is devised specifically for deepfake detection.

3. Evaluation is reasonably thorough and acceptable results are claimed.


Weaknesses：

1. Most of test time training/adaption works, such as Tent [1], emphasize its advantage of computationally efﬁciency and capability of fast adaptation. Comparatively, it seems very time-consuming in the proposed Online Test-Time Training as the proposed Generating pseudo-training samples involves facial alignment, mask refinement and facial region blending. It is hard to be carried out in an ‘online’ manner and more likely to be an offline pipeline.

2. In line 136, the reviewer is not clear about how many x_r are sampled. It is likely to cause the problem of domain imbalance (domain here means template and synthesis domains) if many template images are sampled and only one synthesized image is generated.

3. Offline Meta-training is trivial and directly borrows the ideas of MAML.

4. Missing some SOTA test time training/adaption baselines, such as Tent [1].

[1] Tent: Fully Test-Time Adaptation by Entropy Minimization. ICLR 2021.

---

> ### Author Response · Authors · 2022-08-02
> **Discussion on time consumption, details in pseudo sample generating process, contribution of the offline training stages, and extened evaluations**
>
> We thank the reviewer for the comments, and we answer the raised questions below.
>
> * **1. Time consumption of OST**
>
> The average running time of OST is 0.065 seconds for an image with a resolution of $256 \times 256$, and the average running time of the pseudo training sample generating process is 0.074 seconds as some of the blending steps are processed with a CPU. Note that the facial alignment, mask refinement, and facial region blending steps are not always required during inference. When using the learning-based generating process is selected, we can directly use the test and template samples as inputs and output a pseudo training sample. In this case, the pseudo training sample generating process is 0.021 seconds, which is less than the inference time. However, it is noteworthy to point out that the deepfake detection community now focuses more on  accuracy over speed. Please also see our response to Reviewer QiuQ and Section B in the appendix of our revised manuscript for detailed resource comparisons.
>
> * **2. Numbers of sampled $x_r$**
>
> For every test sample $x_e$, we randomly select one template image $x_r$ from the training set and generate a pseudo training sample $x_o^f$. In other words, one synthesized image corresponds to one template image, and the numbers of the template and pseudo training sample will always be equal.
>
> * **3. Contribution of the offline meta-learning step**
>
> The offline meta-learning step is used to learn a good initialization for the test-time adaptation step. It mimics the training steps of OST. First, we randomly select a template image for the current training sample and generate two pseudo samples. Then, we use one pair of the template and pseudo samples as a support set for inner update and use another pair of training and pseudo samples as a query set for meta update. The overall pipeline is directly borrowed from MAML. We also test our method without using the MAML framework, and the model performs on par with it. We use MAML for our method to enable fast adaptation during inference. Please also refer to Section 5.3 in our manuscript for detailed descriptions.
>
> * **4. Compare with TENT**
>
> We compare with TENT by training the model on the four data from the FF++ dataset, and test it on the DFDC, DFD, and DF1.0 datasets. Results are listed below. We note that TENT performs less effectively against our method in most cases. One possible reason is that the small entropy regularization in TENT [46] may not be as effective in the binary classification task as it is in the multi-class classification task.
>
> ||| DF|||F2F|||FS|||NT|||
> |------|---|---|---|----|---|-----|----|---|-----|----|---|-----|----|
> |Method|DFDC|DFD|DF1.0|DFDC|DFD|DF1.0|DFDC|DFD|DF1.0|DFDC| DFD|DF1.0|Avg.
> |TENT|0.749|0.851|0.915|0.707|0.826|0.916|0.725|0.752|0.915|0.748|0.840|0.886|0.819|
> |Ours |0.757|0.869|0.938|0.798|0.880|0.947|0.802|0.824|0.909|0.752|0.841|0.929|0.854|
>
> * **5. Evaluations with multiple gradient descent steps**
>
> To study if more updating steps can improve the detection, we conduct ablation studies by using different gradient descent steps during evaluation. The model is trained on the FF++ dataset, and it is evaluated on the DFDC, DFD, and DF1.0 datasets. As shown below, the performances of using more gradient descent steps do not visibly improve detection accuracy while the time consumptions (TC) are proportional to the number of gradient descent steps. In addition, using more gradient descent steps also requires much more memory costs in a MAML-based framework. For those reasons, we use only one gradient descent step in our method as a compromise between efficiency and accuracy.
>
> |||DF|||F2F|||FS|||NT||||
> |------|---|---|---|----|---|-----|----|---|-----|----|---|-----|----|----|
> |Steps|DFDC|DFD|DF1.0|DFDC|DFD|DF1.0|DFDC|DFD|DF1.0|DFDC|DFD|DF1.0|Avg.|TC(s)|
> |1 update|0.757|0.869|0.938|0.798|0.880|0.947|0.802|0.824|0.909|0.752|0.841|0.929|0.854|0.065|
> |2 updates|0.759|0.902|0.927|0.794|0.892|0.942|0.797|0.830|0.922|0.758|0.836|0.947|0.859|0.121|
> |3 updates|0.783|0.860|0.942|0.804|0.906|0.934|0.801|0.840|0.929|0.747|0.843|0.948|0.861|0.181|

---

> ### Author Response · Authors · 2022-08-06
> **Sincerely Look Forward to Your Feedback!**
>
> Dear reviewer DZTC,
>
> Thanks again for your insightful suggestions and comments. As the deadline for discussion is approaching, we are glad to provide any additional clarifications that you may need.
>
> We have carefully studied your comments and added additional experiments and analyses in our previous responses to address your concerns. We genuinely hope you could kindly check our responses.
>
> We hope that the new experiments and additional explanations have convinced you of the merits of our work. Please do not hesitate to contact us if there are other clarifications or experiments we can offer.
>
> Thank you for your time again!
>
> Best wishes,
>
> Authors

---

### Official Review · Reviewer_ajm1 · 2022-07-10

**Rating:** 5
**Confidence:** 3
**Soundness:** 2 fair
**Presentation:** 3 good
**Contribution:** 2 fair

**Summary:**

The major contributions of this paper is:
1) This work proposes a simple **One-Shot Test-Time training (OST)** framework to improve detection of Out-of-Distribution face forgery detection. OST obtains noticeable improvements in OOD detection.


**Questions:**

Please see Weaknesses section above for a list of questions.

**Limitations:**

The authors have discussed the limitations / ethic statement in Section 6 (Main Paper).

**Strengths And Weaknesses:**

**Strengths:**
1) This paper is written well. It is easy to follow.
2) The improvements in generalization are interesting and useful to the face forgery detection community.


**Weaknesses:**
1) Although the method gives improvement and I appreciate the authors' comprehensive experiments, these improvements are not surprising. I agree that it is different from other existing methods, but I feel that the contributions are limited. I.e.: Only Line 1 in Algorithm 1 is proposed in this work, Lines 2-4 are from existing meta-learning works.

2) For detecting CNN-generated images, there is a popular work that shows its possible to create a universal detector that generalizes to detecting CNN-generated images from unseen GANs with different architectures, datasets and loss functions (See [1]). This generalization is obtained by a simple threshold calibration with no gradient updates (See appendix Table 5 [1]) of this universal detector. Can the authors include some discussion / benchmark regarding this generalization [1].

3) Can the authors clarify as to why OST cannot be applied to GAN-synthesized images? I.e.:  The universal detector [1] generalizes to face forgery detection (See Table 2 for Cross-generator generalization results) although only trained using CNN-generated images.


Overall this is an interesting paper. In my opinion, the weaknesses of this paper outweigh the strengths. I’m willing to change my opinion based on the rebuttal.


=====

[1] Wang, S. Y., Wang, O., Zhang, R., Owens, A., & Efros, A. A. (2020). CNN-generated images are surprisingly easy to spot... for now. In Proceedings of the IEEE/CVF conference on computer vision and pattern recognition (pp. 8695-8704).

---

> ### Author Response · Authors · 2022-08-02
> **Discussions on the main contribution, generalization with threshold calibration, and the limitation in totally GAN-synthesized images**
>
> We thank the reviewer for the comments, and we answer the raised questions below.
>
> * **1. Main contribution**
> We want to emphasize that our main contribution is to design a test-time training objective specially for the generalizable deepfake detection task. We then apply the existing meta-learning framework, i.e. MAML, to our main idea for better test-time training speed. In other words, the novelty our method does not lie in the meta-learning algorithm or the test-time training principle but in their applications and special development for deepfake detection.
>
>
> * **2. Generalizing with calibrated threshold**
>
> Following the same setting [a], we calibrate the model with a randomly selected pristine and deepfake pair from the test dataset. We first augment the image pair 128 times to obtain a small calibration set. Then the small calibration set is passed into our detection model to get the logits, which are then fitted by logistic regression. Similarly, we take the weight and bias learned from the logistic regression to adjust the output of the evaluated models.  All the models are trained on the FF++ dataset, and the original and calibrated accuracies on three different benchmarks are shown below (original / calibrated). We observe that the calibration does not improve the accuracy in all cases. This phenomenon is consistent with the observations in [a].
>
> |Original & Calibrated| DFDC  | DFD | DF1.0 |
> |---| --- | --- | --- |
> |Xception|0.671 & 0.704 |0.693 & 0.688 |0.543 & 0.622|
> |Face X-ray|0.659 & 0.674|0.653 & 0.649|0.625 & 0.684|
> |F3Net|0.657 & 0.691 | &0.604 & 0.659 | 0.707 & 0.693|
> |RFM|0.731 & 0.702 |0.783 & 0.796 |0.625 & 0.649 |
> |SRM|0.696 & 0.688| 0.695 & 0.714 |0.684 & 0.677|
> |Ours|0.714 & 0.703|0.831 & 0.858 | 0.903 & 0.881|
>
> * **3. Limitation in totally GAN-synthesized images**
>
> In fact, a universal detector trained on GAN-synthesized images does not perform well on deepfake images. Table 5 from the appendix of [a] shows that the accuracy of the detector evaluated on the FF++ dataset is around 50%, much lower than that evaluated on other GAN-synthesized images. These experiments indicate that we may need a different set of clues than those used for detecting GAN-synthesized images to detect deepfake images. This also explains many recent works only study the detection of deepfake images [23, 30, 52, 18, 37, 4, 34, 53, 12, 19, 17]. Our OST follows this line of research. More specifically, same as existing deepfake detection arts, OST is developed based on the assumption that a deepfake contains contents from different sources, and contents in a real image are from only one source. That is also why the pseudo training sample is regarded as fake even if they are blended with two real images. However, for the totally GAN-synthesized situation, the fake images contain only one source, and images without GAN patterns, even if they are synthesized by two real images, are still considered real. This setting contradicts the assumption in current deepfake detection, and it certainly requires more effort to detect them perfectly in one unified framework.
>
> [a] CNN-generated images are surprisingly easy to spot... for now. In CVPR 2020.

---

> ### Author Response · Authors · 2022-08-06
> **Sincerely Look Forward to Your Feedback!**
>
> Dear reviewer ajm1,
>
> Thanks again for your insightful suggestions and comments. As the deadline for discussion is approaching, we are glad to provide any additional clarifications that you may need.
>
> We have carefully studied your comments and added additional experiments in our previous responses to address your concerns. We genuinely hope you could kindly check our responses.
>
> We hope that the new experiments and additional explanations have convinced you of the merits of our work. Please do not hesitate to contact us if there are other clarifications or experiments we can offer.
>
> Thank you for your time again!
>
> Best wishes,
>
> Authors

---

> > ### Comment · Reviewer_ajm1 · 2022-08-09
> > **Reply**
> >
> > Thank you authors for the great effort on the rebuttal. Authors have addressed my concerns to some extent.
> >
> > **In the revised version, please consider including a short description to compare against GAN-synthesized image detectors ([1, 2, 3]) to accurately convey the scope of your work.**
> >
> > **Although I still stand by my initial review regarding limited technical novelty ( See weakness (1) ), given that the proposed method could be useful in face-forgery detection applications, I will increase my recommendation accordingly.**
> >
> > [1] Wang, S. Y., Wang, O., Zhang, R., Owens, A., & Efros, A. A. (2020). CNN-generated images are surprisingly easy to spot... for now. In Proceedings of the IEEE/CVF conference on computer vision and pattern recognition (pp. 8695-8704).
> >
> > [2] Dzanic, T., Shah, K., & Witherden, F. (2020). Fourier spectrum discrepancies in deep network generated images. Advances in neural information processing systems, 33, 3022-3032.
> >
> > [3] Chandrasegaran et al., 2021: "A closer look at fourier spectrum discrepancies for cnn-generated images detection." Proceedings of the IEEE/CVF Conference on Computer Vision and Pattern Recognition. 2021.

---

> > > ### Author Response · Authors · 2022-08-09
> > > **Thanks very much for the reply!**
> > >
> > > Many thanks for the valuable suggestions. These works will be included and discussed in our future version.

---

### Official Review · Reviewer_QiuQ · 2022-07-11

**Rating:** 8
**Confidence:** 5
**Soundness:** 4 excellent
**Presentation:** 3 good
**Contribution:** 4 excellent

**Summary:**

This paper presents an approach for generalizable Deepfake detection using a recently proposed one-shot test-time training strategy and a combination of meta learning. The approach is simple and a straightforward extension of the recently proposed test-time training framework, where a test-data sample itself is used to create a pseudo-training set, and the model parameters are updated. Experiments are presented on various datasets under different experimental settings.


**Questions:**

The paper uses a face detector method (DLIB) to extract faces. There is not much information on how accurate this method is. The paper appears to assume that the face detection accuracy is 100%. Since face detection is critical to the proposed method, are there scenarios when the face detection algorithm can fail? Some discussion on this will help.

The paper mentions that the input face images are resized to a dimension of 256x256. It will be good to have a discussion on why this dimension is chosen, and if choosing a higher or lower dimension will have an impact on the performance of the proposed method.


**Limitations:**

Lack of good visual examples is a big limitation (having which could have ended up in a higher rating).

The paper uses ACC and AUC as evaluation metrics. It will be helpful to know how the proposed method works on both pristine and Deepfake images. Metrics like True positives, False positives, True Negatives and False Negatives could help here. Since only ACC and AUC are provided, this does not reflect how well this algorithm performs separately on pristine and Deepfake images.

There is not much discussion on the computational complexity and time complexity of the proposed approach. Since the paper uses test-time training, a good practical system should also discuss the time and computational complexity at the test stage.


**Strengths And Weaknesses:**

Strengths

This paper directly extends the test-time training (TTT) method from Efros’ group, to the Deepfake detection problem, and shows that this can be used for better generalization. Since the TTT method is recent, and has been well received, the authors have cleverly “struck the iron, when it’s hot” and have applied this recently proposed “hot” method to Deepfake detection.

The paper is clearly well and the methods are well explained, for the most part.

Weakness

Visual examples are missing. Though the authors have done a good job in extending a recently proposed well received method to Deepfake detection, it doesn’t make sense why the authors have not included any good visual examples. Other than Figure 2 (which hardly illustrates the method), there are no visual images. Since the method takes a test image and blends with images from the training set using different blending methods, these can be easily visualized and thus provide better insights.

A few experimental scenarios and details are missing.

---

> ### Author Response · Authors · 2022-08-02
> **Inluding visual examples, dicussion on the situation when DLIB fails, and extented evaluations**
>
> We thank the reviewer for the comments, and we answer the raised questions below.
>
> * **1. Visual examples**
>
> We include a visual example to demonstrate the generating process of the pseudo training sample. Please refer to Section A and Figure 4 in the appendix of the revised manuscript for a detailed description.
>
> * **2. DLIB face recognition fails**
>
> DLIB is a commonly used machine learning toolkit. As a subset, the face detection function in DLIB can achieve up to 99.38% accuracy on the Labeled Faces in the Wild benchmark. Indeed, there are circumstances that DLIB may fail, and in which cases, our OST  method may also fail since the pseudo training sample generating process requires the landmarks of the faces. We will include it as a limitation in our future version.
>
> * **3. Experiments on images with different resolutions**
>
> Following the settings in previous works [23,30], we use images with the resolution of $256\times 256$ in our method. To evaluate if the resolution can also influence the generalizability of the detector, we conduct ablation studies by using samples with different resolutions. Results are listed below. We observe that the differences between the three different resolutions are rather small (less than 1\% on average), indicating that image resolution is not a major influential factor.
>
> ||| DF|||F2F|||FS|||NT|||
> |------|---|---|---|----|---|-----|----|---|-----|----|---|-----|----|
> |Resolution|DFDC|DFD|DF1.0|DFDC|DFD|DF1.0|DFDC|DFD|DF1.0|DFDC| DFD|DF1.0|Avg.
> |200 $\times$ 200|0.741|0.855|0.962|0.782|0.862|0.959|0.790|0.794|0.947|0.713|0.812|0.937|0.846|
> |320 $\times$ 320|0.755|0.916|0.937|0.721|0.858|0.948|0.843|0.801|0.939|0.760|0.823|0.931|0.853|
> |256 $\times$ 256|0.757|0.869|0.938|0.798|0.880|0.947|0.802|0.824|0.909|0.752|0.841|0.929|0.854|
>
> * **4. Evaluations with different metrics**
>
> To evaluate how the compared methods perform on pristine and deepfake separately, we also report true positive (TP), true negative (TN), false negative (FN), false positive (FP), and true negative rate (TNR) for them. Results are listed below. We observe that the TNR of our method is much larger than other methods, indicating that our method is more likely to correctly detect a given deepfake image.
>
> |Dataset| DFDC||||DFD||||DF1.0|||||
> |------|---|---|---|----|---|---|---|----|---|---|---|----|---|
> |Metric|TP|TN|FN|FP|TP|TN|FN|FP|TP|TN|FN|FP|Avg. TNR|
> |Xception|1410 |700|123 |913|783|5375|138|2588|8857|2064|1193|7986|0.415|
> |Face X-ray |1288|784|245 |829 |814 |4984 |107 |2979 |9435 |3118 |615 |6932 |0.453|
> |F3Net|1387|679|146|934 |902 |4465 |19 |3498 |9338 |4880 |712 |5170 |0.511|
> |RFM|1109|1192|424|421 |763 |6191 |158 |1772 |9731 |2835 |319 |7215 |0.521|
> |SRM|1348|841|185|772 |874 |5302 |47 |2661 |10037 |3712 |13 |6338|0.502|
> |Ours|1032|1214|501|399|781 |6602 |140 |1361 |8336 |9816 |1714 |234 |0.898|
>
> * **5. Time consumption and computational complexities evaluations**
>
> To comprehensively evaluate the proposed method, we provide the time consumption (TC) and computational complexity (CC) comparisons below. All the compared methods are evaluated on the same device using a $256 \times 256$ image, and they are all implemented with the Xception backbone. Thus the computational complexities are nearly the same for most models except for SRM which uses a dual branch network architecture. Because our model includes two forward and one backward operations during inference, thus the corresponding computational complexity is more than others. Meanwhile, our method involves the generation of pseudo training samples that mostly use the CPU for the task except for when using the learning-based generating method (0.074 seconds on average for the pseudo training sample generation process). Thus, it requires more running time than others but also at an acceptable speed. Moreover, it is noteworthy to point out that the current focus of deepfake detection is still on the detection accuracy rather than speed. This might be because deepfake detection system might not be speed-sensitive in many scenarios. Certainly, methods can be further developed in the future, e.g., using distillation or approximation of online update, to further accelerate our method.
>
> ||Xception|Face X-ray|F3Net|RFM|SRM|OST|
> |-----| --- | --- | --- |--- | --- | --- |
> |TC (s) | 0.015|0.017|0.019|0.015|0.037|0.062+0.074|
> |CC (MACs(G))| 6.01|6.01|6.05|6.01|13.81|18.03|

---

### Author Response · Authors · 2022-08-02
**General Response**

We sincerely appreciate all reviewers' efforts in reviewing our paper and giving insightful comments and valuable suggestions. We are glad to find that the reviewers generally acknowledge the following novelty and contributions of our work.

*  **Main contribution.**  We introduce a test-time training paradigm specially designed for the deepfake detection task. Specifically, for each test image, we can use it to synthesize a pseudo training sample with existing deepfake generating techniques. Because the label of the pseudo sample is known (i.e. fake), we thus can use it to update the detector during inference.

As suggested by the reviewers, we would like to include the following contents in our revised manuscript to further improve our paper. We summarize the major revision as follows. Our detailed responses can be found in the following response sections to the reviewers.

*  **Visual examples.**  We add visual examples to better illustrate the pseudo training sample generation pipeline and visualization of the embedded representations.

*  **Resources usages.** We include time consumption and computational complexity comparisons for all the compared arts.

*  **Ablation studies.**  We include more ablation studies regarding using multiple gradient descents in the online adapting step, evaluations on images with different resolutions, and results by using the calibrated threshold.

*  **More comparisons.** Comparisons with more methods including those based on meta learning and test-time training (i.e. MT3 [5] and TENT [46]), and comparison using other evaluation metrics such as true positive and false positive.

Please also refer to the appendix in our revised paper for more detailed descriptions.

---

### Meta-Review · Area_Chair_9Rwn · 2022-08-27

**Recommendation:** Accept
**Confidence:** Certain

**Metareview:**

The reviewers unanimously accept the paper, so is the final proposal.

**Award:**

No

---

### Decision · Program_Chairs · 2022-09-14

Accept